# A Review of Medicinal Plants Used in the Management of Microbial Infections in Angola

**DOI:** 10.3390/plants13212991

**Published:** 2024-10-26

**Authors:** Dorcas Tlhapi, Ntsoaki Malebo, Idah Tichaidza Manduna, Thea Lautenschläger, Monizi Mawunu

**Affiliations:** 1Centre for Applied Food Sustainability and Biotechnology, Faculty of Health and Environmental Sciences, Central University of Technology, Bloemfontein 9300, South Africa; imanduna@cut.ac.za; 2Centre for Innovation in Learning and Teaching, Central University of Technology, Bloemfontein 9300, South Africa; nmalebo@cut.ac.za; 3Botanical Garden Hamburg, University of Hamburg, 22609 Hamburg, Germany; 4Department of Agronomy, Polytechnic Institute, Kimpa Vita University, Luanda P.O. Box 77, Angola; m.mawunu2000@gmail.com

**Keywords:** Angolan medicinal plants, antimicrobial activity, antibacterial activity, botanical description, distribution, taxonomy, ethnobotany, phytochemistry, pharmacology

## Abstract

The use of medicinal plants in the management of microbial infections is significant to the health of the indigenous people in many Angolan communities. The present study provides a comprehensive overview of medicinal plants used for the management of microbial infections in Angola. Relevant information was extracted from research articles published and associated with the use of medicinal plants in the management of microbial infections in Angola (from January 1976 to November 2023). Data or information were gathered from the literature sourced from Wiley Online, SciFinder, Google Scholar, Web of Science, Scopus, ScienceDirect, BMC, Elsevier, SpringerLink, PubMed, books, journals and published M.Sc. and Ph.D. thesis. A total of 27 plant species, representing 19 families, were recorded in this study. Hypericaceae (11%), Lamiaceae (11%), Malvaceae (11%), Phyllanthaceae (11%), Fabaceae (16%) and Rubiaceae (16%) were the most predominant families. The leaves are the most used parts (96%), followed by bark (74%) and root (70%). The data revealed that medicinal plants continue to play significant roles in the management of microbial infections in Angola. In order to explore the benefits of the therapeutic potential of indigenous medicinal plants for diseases related to infections; further scientific research studies are important to produce data on their effectiveness using appropriate test models. This approach might assist with the continuing drive regarding the integration of Angolan traditional medicine within mainstream healthcare systems.

## 1. Introduction

Infectious diseases remain a crucial problem in global public health and are one of the leading causes of death worldwide [1]. According to the literature reports, these ailments are responsible for 17 million deaths per year worldwide [2,3]. Treatment of microbial infections is becoming a challenge due to the emergence of bacterial pathogens resistant to all antimicrobial drugs [4]. Therefore, there is a need to discover new alternative antimicrobial constituents with improved activity. Traditional herbal medicine has played an essential role as antibiotics in various developing countries [5]. Medicinal plants have long been considered a natural source of novel remedies of high value to mankind [6]. Natural medicinal plants are proven to have beneficial effects and have been used globally for healing various diseases [7]. Medicinal plants are a rich source of phytochemicals and have been used for their antiviral, antifungal, and antibacterial activities for many years [8]. Plants that exhibited the presence of antibacterial components provided data, which led to the development of novel antimicrobial medications [9]. About 80% of people worldwide depend on traditional plants or herbs for their primary healthcare [10]. Fortunately, some plant products, either in the form of raw materials or isolated compounds, have been tested for possible antimicrobial and antibacterial therapy in Africa, but the number of plants and isolated bioactive compounds with potential antimicrobial properties is very limited, and their antimicrobial and antibacterial activities have not yet been scientifically validated [11]. The present study is a review of medicinal plants used to treat microbial and bacterial infections in Angola. The information presented in this review may be used as a guide to formulate new and effective medicinal drugs used to treat microbial infections.

### 1.1. Traditional Medicine in Angola

Angola is the largest country in Southern Africa, with an estimated population of over 37.2 million, according to the Angola Population (2024)—Worldometer [12]. The Republic of Angola (Figure 1) covers an area of 1,246,700 km^2^ (481,400 square miles) in Southwest Central Africa. Its western boundary is 1650 km along the Atlantic Ocean and it is bordered by Namibia in the south, the Democratic Republic of Congo in the north and northeast, and Zambia in the east [13]. Furthermore, Angola is the second richest country with respect to endemic plants in continental Africa [14]. The total number of vascular plants in Angola is approximately 6850 species, of which 14.8% are endemic and frequently used in traditional medicine [15,16]. The rich flora of Angola suggests enormous potential for the discovery of new secondary plant metabolites with therapeutic value. There are only a few reports and studies documenting the traditional use of plants in different regions of Angola [17,18,19,20,21,22]. Therefore, there is a need to conduct ethnobotanical, ethnomedicinal and ethnopharmacological studies, document the diversity of medicinal plants and related ethnomedicinal knowledge of traditional healers or herbalists in Angola.

A wealth of Angolan ethnic knowledge has been transferred from generation to generation [23,24]. Many Angolans rely on plant-based medicinal treatment for primary healthcare [23,24,25]. This is because Angola’s healthcare system is poor, compared to other sub-Saharan countries in Africa. The lack of essential public health facilities, especially in rural areas, is a major challenge resulting in the importance of traditional practitioners and herbal medicines. In Angola, as in several other countries, the use of plants is an important cultural phenomenon, with some ailments recognised as specifically requiring traditional medicine [22]. Different plant parts are frequently used to treat, manage and control various diseases or sicknesses, such as acute respiratory, cholera, diarrhoea, malaria, tetanus and yellow fever, which affect their patients [22,26]. This is because most medicinal plants are inexpensive, accessible and easy to prepare [27].

### 1.2. The Burden of Antimicrobial Resistance

Antimicrobial resistance (AMR) is a major global health threat and directly impacts economic growth worldwide [28]. However, low- and middle-income countries, especially in Africa, bear the biggest burden of the negative outcomes of antimicrobial resistance [29]. Antimicrobial resistance occurs when microorganisms (bacteria, viruses, fungi and parasites) grow or develop in the presence of drugs invented to prevent or kill them [30]. This results in therapeutic failure, increased disease spread, severe sickness, disability, and high mortality rates, which negatively influence the global management and control of infectious diseases [31]. Infections caused by antimicrobial-resistant microorganisms are reported to result in approximately 700,000 deaths worldwide annually [31]. The higher burden on low- and middle-income countries, including Africa, is attributed to poor water sanitation, less access to quality antibiotics, insufficient water supply, less access to the best public health facilities, and travelling [32,33,34]. According to estimates, 5.00 million people died in 2019 from antimicrobial-resistant-related causes, of which 1.3 million deaths were directly associated with bacterial AMR globally [31]. In the African continent, over 1.05 million deaths were associated with antimicrobial resistance, and 250,000 deaths were due to antimicrobial resistance [35]. The number of deaths associated with antimicrobial resistance in Africa is greater than those caused by HIV/AIDS (639,554), malaria (594,348), and tuberculosis (424,000), marking a pivotal shift in the health problems facing the African continent [35]. Four major pathogenic microorganisms (*Escherichia coli*, *Klebsiella pneumoniae*, *Staphylococcus aureus*, and *Streptococcus pneumoniae*) contributed to more than 100,000 deaths in the WHO African region [35]. Furthermore, in Angola, 5300 deaths were due to AMR and 22,700 deaths were associated with antimicrobial resistance [36]. The number of AMR deaths in Angola is greater than deaths from malaria, neglected tropical diseases, neoplasms, enteric infections, sexually transmitted infections, HIV/AIDS, and neonatal and maternal disorders [36]. The five leading pathogens for deaths associated with AMR in Angola were *Group B Streptococcus* (1600), *Staphylococcus aureus* (2200), *Escherichia coli* (3100), *Klebsiella pneumoniae* (3800), and *Streptococcus pneumoniae* (4800) [36]. More than a few antibiotics such as Carbapenem (such as imipenem and meropenem), Fluoroquinolone (such as ciprofloxacin), Methicillin, Penicillins (such as ampicillin), Vancomycin, and third-generation cephalosporin (such as ceftriaxone and cefotaxime) widely used globally for treating *Enterobacteriaceae*, *Escherichia coli*, *Group B Streptococcus*, *Pseudomonas aeruginosa*, *Staphylococcus aureus*, *Klebsiella pneumoniae* and *Streptococcus pneumoniae* strains, respectively, have encountered problems and challenges in their efficacy due to antimicrobial resistance [37,38]. According to the World Health Organization [39], drug-resistant diseases could cause 10 million deaths each year by 2050 and damage the global economy significantly [39]. Furthermore, WHO predicts that antimicrobial resistance will cost an additional USD 1 trillion in healthcare by 2050 if sufficient measures are not fulfilled [40]. Unfortunately, the burden is estimated to be higher in Sub-Saharan Africa, and this calls for necessary measures and interventions to limit the threat [28,40]. Thus, future strategies to prevent AMR must focus on fostering international collaboration with high-income countries to build better health facilities and laboratories in countries that are faced with significant infrastructural challenges; train more laboratory workers or scientists and offer them a reasonable remuneration; educate the public about improving the practice of infection prevention and control (IPC) measures at food handling industries or establishments, home and healthcare facilities; increase the healthcare workers; and increase the public’s knowledge of antibiotics and antimicrobial resistance [41].

## 2. Results and Discussion

### 2.1. Medicinal Plants Used for Microbial Infections in Angola

This review highlights 27 plant species representing 19 different plant families that are widespread in all areas of Angola. The leaves are the most used parts (96%), followed by bark (74%) and root (70%). The predominant families are Hypericaceae (11%), Lamiaceae (11%), Malvaceae (11%), Phyllanthaceae (11%), Fabaceae (16%) and Rubiaceae (16%). Information or data about the medicinal plants presented include taxonomic names, family, plant part(s) used and traditional uses (Table 1).

#### 2.1.1. *Acanthus montanus* (Nees) T. Anderson

*Acanthus monthanus* is a shrub widely distributed in Angola, Chad, Zambia, Sierra Leone and Benin [42]. It belongs to the Acanthaceae family which consists of approximately ~2500 species in ~250 genera [43]. *A. monthanus* is used to treat many diseases like cystitis, pains, endometritis, urogenital infections, aches, leucorrhoea, urethral pain, urinary disease and furuncle [44]. Numerous pharmacological properties of this plant including antipyretic, spasmolytic, anti-inflammatory, antimicrobial, and analgesic have been well-documented [45,46,47]. Okoli et al. (2008) showed that the aqueous root extract of *Acanthus montanus* had antimicrobial activities at concentrations of 50 and 100 mg/mL, with zone of inhibition diameters of 10.0 ± 0.32 and 10.0 ± 0.27; and 19.3 ± 0.18 and 15.0 ± 0.32 against *Staphylococcus aureus* and *Pseudomonas aeruginosa* strains, respectively [48]. In addition, Okoli et al. (2008) indicated that the antimicrobial efficacy of the aqueous root extracts of *Acanthus montanus* may be due to the presence of alkaloids, carbohydrates, saponins, glycosides, and terpenoids [48]. Medicinal plants that have alkaloids and glycosides are claimed to exhibit antimicrobial effects [49,50].

#### 2.1.2. *Abelmoschus esculentus* (L.) Moench

*Abelmoschus esculentus* (“Lady’s Finger”), also known as Okra worldwide, belongs to the family Malvaceae, estimated to contain ~4224 species in ~244 genera [51]. It has various reported biological properties like antidiabetic, antimicrobial, antioxidant, and anti-inflammatory activity [52,53]. These properties are due to the carotene, chlorophyll and phenolic compounds found in *A. esculentus* [53]. *A. esculentus* can be used to treat depression, ulcers, sore throats, pulmonary inflammations, and gastrointestinal irritations [54]. Islam (2019) showed that essential oils: cyclohexanol, citral, β-sitosterol, 3β-galactoside, p-tolualdehyde and α-terpenylacetate isolated from the seeds and pods of *Abelmoschus esculentus* have antibacterial activity [55]. In another study, fresh water and lyophilised extracts of the *A. esculentus* (Okra) pods were effective against *Mycobacterium* sp., *Staphylococcus aureus*, *Xanthobacter* Py2, *Escherichia coli*, *Rhodococcus opacus*, *Rhodococcus erythropolis*, and *Mycolicibacterium aurum* at a concentration of 97.7 mg/mL. The identified saturated fatty acids (palmitic and stearic acids) may be responsible for the antibacterial activity of the fresh water and lyophilised extracts [56].

#### 2.1.3. *Aframomum alboviolaceum* (Ridl.) K.Schum.

*Aframomum alboviolaceum* is part of the family Zingiberaceae, comprised of ~52 genera and more than 1300 species; extensively distributed in Sierra Leone to Sudan, south to Angola, Malawi, Zambia, and Mozambique [57]. All plant parts are used to cure headache, cough, fever, amoebic dysentery, haemorrhoids, gastritis, myoma, pruritis, hypertension, filarial, malaria, pruritus, and gastritis [58,59,60]. Ethnopharmacological studies have shown antihelminthic, antimalarial, antiparasitic, antioxidant, antimicrobial, anti-sickling, and anticancer properties from *Afromomum alboviolaceum* [58,59,60]. Phytochemical evaluation of *A. alboviolaceum* leaves using thin layer chromatography revealed various metabolites such as iridoids, anthocyanins, tannins, coumarins, phenolic acids, flavonoids, and anthraquinones in this species [61]. It has also been shown that *A. alboviolaceum* contains alkaloids, anthocyanines, and triterpenes [62]. Olagoke and Amusat (2019) showed that volatile compounds present in the leaves of *A. alboviolaceum* have antimicrobial potential against *Staphylococcus aureus*, *Pseudomonas aeruginosa*, *Escherichia coli*, *Candida albicans*, *Bacillus subtilis*, and *Aspergillus niger* at 100%, 50% and 25%, as well as moderate inhibition at 12.5 and 6.25% [63].

#### 2.1.4. *Alchornea cordifolia* (Schumach. & Thonn.) Müll.Arg.

*Alchornea cordifolia* is a small tree in the Euphorbiaceae family, containing ~6745 species in ~218 genera [64,65]. It is distributed throughout tropical Africa [64,66]. The plant is used in Africa to treat venereal diseases, malaria, diarrhoea, wounds, cuts, sores, coughs, colds, eye problems, pigmentation problems, gastrointestinal, headaches, rheumatic pains, urinary disorders, as well as inflammatory disorders, and fungal, parasitic, bacterial, and parasitic disorders [67,68,69,70]. Some reports on the pharmacological activities of *A. cordifolia* have confirmed anti-inflammatory, antiviral, anti-diarrhoeal, hepaprotective, and antidiabetic activity [71,72,73,74,75]. Furthermore, several antimicrobial tests of the plant showed its efficacy against a wide variety of microorganisms such as skin, gastrointestinal, urinary tract and respiratory pathogens [65,76,77,78], therefore supporting the medicinal use of the plant species for the management of such diseases. The biological activities of *Alchornea cordifolia* were due to terpenoids, flavonoids, tannins, steroid glycosides, phenolic acids, saponins, imidazopyrimidine, alkaloids, and fatty acids, carbohydrates [79]. Essien et al. (2015) showed that volatile compounds such as 28.0% benzaldehyde, 25.3% methyl salicylate, 21.4% citronellol, 15.5% β-caryophyllene, 7.4% α-phellandrene, 5.7% terpinolene, 5.5% 1,8-cineole, and 5.3% (*E*,*E*)-α-farnesene in the essential oil of *A. cordifolia* fruits contributed to the antifungal potential against *Staphylococcus aureus*, *Aspergillus niger* and *Bacillus cereus*, with MIC values of 78 μg/mL, 156 μg/mL and 156 μg/mL, respectively [64].

#### 2.1.5. *Aloe buettneri* A. Berger

*Aloe buettneri* (Asphodelaceae, which includes about ~900 known species and ~40 genera) is a flowering succulent plant species that has been traditionally utilised in West Africa to treat wounds, vitiligo, malaria, rheumatism, insect bites, burns and worm sores [80]. This plant has a wide variety of chemical constituents, including polyphenols, tannins, flavonoids, sterols, terpenoids, saponins, alkaloids and carbohydrates [81]. However, the bioactive compounds and their bioactivities are not well known or described. The extracts of *A. buettneri* have a wide variety of pharmacological activities, such as antioxidant, gastric antisecretory, anti-ulcer, and anti-inflammatory [82,83]. Kombate et al. (2022) found that the ethanol–water mixture (5:5. *v*/*v*) leaf extract had antimicrobial activity, with inhibition zones of 18.20 ± 0.10 and 14.24 ± 0.17 mm against *Cutibacterium acnes* and *Pseudomonas aeruginosa* at a concentration of 250 mg/mL [81]. Kombate et al. (2022) also indicated that *A. buettneri* gave the best activity with 31.25 mg/mL for minimum microbicidal concentrations and 15.625 mg/mL for MICs (minimum inhibitory concentrations) against *Klebsiella pneumoniae* [81]. Their phytochemical screening results showed the presence of saponosides, triterpenes, carbohydrates, tannins, flavonoids, polyphenols, and alkaloids in the ethanol–water mixture (5:5. *v*/*v*) leaf extract of *A. buettneri,* which explained the antimicrobial activities [81]. A few literature studies have shown that saponins, tannins, flavonoids, triterpenoids and polyphenolic compounds are also known to have antimicrobial activities [84,85,86]. Therefore, the secondary metabolites found in the ethanol–water mixture (5:5. *v*/*v*) leaf extract of *A. buettneri* may have acted on their own or in synergy to inhibit the tested pathogenic microorganisms.

#### 2.1.6. *Annona stenophylla* Engl. & Diels

*Annona stenophylla* belongs to the Annonaceae family, which includes about ~108 genera and ~2400 known species. Many people worldwide use *A. stenophylla* to treat coughs, wounds, helminthiasis, cancer, dysentery, diarrhoea, headache, asthma, fever, malaria, dermatitis, mental disorders, diabetes, and peptic ulcers [87,88,89]. *Annona* species are characterised by biological activities such as anti-inflammatory, hypoglycaemic, antimicrobial, anticholinesterase, analgesic, antioxidant, antiparasitic, anticonvulsant, antiplatelet, hepato-protective, anxiolytic, cytotoxicity, and antiproliferative; these activities are usually attributed to the presence of quinones, sesquiterpene lactones, sterols, essential oils, acetogenins, alkaloids, and terpenes [88,90,91]. Munodawafa (2008) and Munodawafa et al. (2013) assessed the antibacterial effects of methanol root and leaf extracts of *A. stenophylla* against *Pseudomonas aeruginosa*, *Escherichia coli*, and *Staphylococcus aureus* [92]. They showed that the root extract was active against *Pseudomonas aeruginosa* with an inhibition zone ranging from 2.8 to 3.0 mm, whereas the leaf extract was active against *Staphylococcus aureus* with an inhibition zone ranging from 1.0 to 1.5 mm. Furthermore, Munodawafa (2008) and Munodawafa et al. (2013) determined the antifungal effects of methanol root and leaf of *A. stenophylla* against *Aspergillus niger* and *Candida albicans* [92]. Their results showed that the root extract of *A. stenophylla* showed activities against pathogenic microorganisms with inhibition zones ranging from 1.5 to 3.8 mm. Plant secondary metabolites: coumarins, alkaloids, cardiac glycosides, anthraquinone derivatives, tannins, saponins, and flavonoids identified in the methanol root and leaf extracts of *A. stenophylla* by phytochemical screening could have contributed separately or in combination to the potent antibacterial and antifungal activities of this plant [92].

#### 2.1.7. *Bridelia ferruginea* Benth.

*Bridelia ferruginea* (Phyllanthaceae) grows in the rainforests or savannahs of Africa, Australia, and Southern Asia [93]. Phyllanthaceae comprises about ~2000 species grouped into ~60 genera. The roots, fruits, leaves and bark of the “woody shrub” are commonly prepared traditionally for managing dysentery, arthritis, cough, rashes, constipation, epilepsy, diarrhoea, diuretic, chronic diabetes, asthma, gout, intestinal disorders, thrush, dysentery, gastralgias, rheumatisms, sexually transmitted diseases, contusion, oral infections, bladder disorders, skin diseases, and anaemia [94]. Biological studies on several *B. ferruginea* extracts support its use as an antipyretic, anthelmintic, antityphoid, antioxidant, analgesic, antimicrobial, anti-inflammatory, antidiabetic, and antiplasmodial properties in different parts of Africa [95,96,97,98,99,100,101,102]. Different chemical compounds reported to be found in *B. ferruginea* extracts include tannins, triterpenes, saponins, alkaloids, flavonoids, cardiac glycosides, phenolics, and phytosterols [103]. A recent study by Afolayan et al. (2019) showed that β-sitosterol glucoside, rutin, quercitrin, myricitrin, isoquercetin, isomericitrin, myricetin, kaempferide-3-*O*-β-D-glucoside, corilagin, vomifoliol, lutein, 6β-hydroxy-(20R)-24-ethylcholest-4,22-dien-3-one, 6β-hydroxy-(20R)-24-ethylcholest-4-en-3-one, oleic acid, stearic acid and palmitic acid were isolated and identified from the methanolic leaf extracts of *B. ferruginea* [104]. Myricitrin was the only compound that showed antibacterial activities against *Escherichia coli* with an IC_50_ value of 1.123 μM. [104]. Irobi et al. (1994) showed that phenols and tannins in *B. ferruginea* might be responsible for the antimicrobial activities against *Candida albicans*, *Escherichia coli*, *Proteus vulgaris*, *Klebsiella pneumoniae*, *Staphylococcus epidermidis*, *Staphylococcus aureus*, *Proteus mirabilis*, *Streptococcus lactis* and *Streptococcus pyogenes* at a concentration of 5 mg/mL [105].

#### 2.1.8. *Canarium schweinfurthii* Engl.

*Canarium schweinfurthii* (Burseraceae (~19 genera and about ~540 species) is a species of a large tree found in tropical Africa that can be used to treat roundworm infections, skin affections, dysentery, eczema, diarrhoea, dysentery, haemorrhoids, venereal diseases, hypertension, malaria, fever, gonorrhoea, chest pains, stomach complaints, pulmonary affections, ulcers, leprosy, coughs and wounds [106,107,108,109]. The plant has numerous critical biological activities, such as antidiabetic, antibacterial, antifungal, analgesic, antiparasitic, anti-inflammatory and antioxidant [110]. This plant has been reported to have several active metabolites such as sterols, fatty acids, saponins, triterpenes, alkaloids, flavonoids, tannins, anthraquinones, polyphenols, and coumarins [111,112,113]. According to Dzotam et al. (2016), *C. schweinfurthii* extracts had antimicrobial activities against *Escherichia coli*, *Enterobacter aerogenes*, *Klebsiella pneumoniae*, *Pseudomona aeruginosa* and *Providencia stuartii*, with minimum inhibitory concentration values that were between 64 to 1024 μg/mL [114]. Qualitative phytochemical analysis showed that the methanol *C. schweinfurthii* extracts contained saponins, sterols, triterpenes, tannins, anthraquinones, flavonoids, polyphenols and alkaloids [114]. Therefore, the antimicrobial effects of this plant could be due to the differences in their chemical composition as well as in the mode of action of their bioactive compounds. Nagawa et al. (2015) revealed that the main compounds found in essential oil of *Canarium schweinfurthii* resin were mostly α-thujene, γ-terpinene, β-pinene, p-cymene, β-phellandrene, α-phellandrene, α-pinene, sabinene, nerolidol octyl acetate, n-octanol, limonene and α-terpineol [115]. However, these compounds could not be tested for their biological activities due to small amounts of the compounds after qualitative chemical analysis. Several research studies reported that α-thujene, γ-terpinene, β-pinene, p-cymene, β-phellandrene, α-phellandrene, α-pinene, sabinene, nerolidol octyl acetate, n-octanol, limonene and α-terpineol were responsible for the antimicrobial activities of several essential oils found in medicinal plants [116,117,118,119,120,121,122]. Consequently, the strong antimicrobial activity of *Canarium schweinfurthii* might be due to the presence of these main compounds found in the essential oils of this plant.

#### 2.1.9. *Chromolaena odorata* (L.) R.M.King & H.Rob.

*Chromolaena odorata* (Asteraceae) is mainly found in tropical and subtropical areas of Texas, Florida, Mexico, Australia, and Asia. Asteraceae is a large family of flowering plants that consists of over ~32,000 known species in over ~1900 genera [123,124]. Various therapeutic and medicinal properties of *C. odorata*, which include antioxidant, anthelmintic, antimicrobial, antimalarial, analgesic, antispasmodic, anti-inflammatory, and antipyretic, have been reported [125,126,127]. It is widely used for malaria, coughs, colds, toothache, stomach problems, diarrhoea, stomach ulcers, wounds, dysentery, skin infections, and bacterial and fungal infections. This plant has many components like saponins, tannins, coumarins, steroids, terpenoids, cardiac glycosides, and flavonoids [128,129]. Atindehou et al. (2013) demonstrate that the cyclohexane, dichloromethane, ethyl acetate and butanol leaf extracts of *C. odorata* had antibacterial activity ranging from 0.156 to 1.25 mg/mL against *Vibrio cholerae*, *Shigella sonnei*, *Salmonella enterica*, and *Klebsiellaoxytoca* microorganisms [130]. The best antibacterial activity was obtained against *V. cholerae* strain with MIC values of 0.156 mg/mL and 0.312 mg/mL for the dichloromethane and butanol extracts, respectively [130]. Atindehou et al. (2013) also characterised two flavonoids: sinensetin (3’,4’,5,6,7-pentamethoxyflavone) and scutellareintetramethyl ether (4’,5,6,7-tetramethoxyflavone) using bioassay-guided isolation by chemical and pharmacological approaches. However, the antibacterial activity of these compounds could not be tested due to insufficient amounts of material after chemical analysis [130]. In addition, other research studies have shown that these two flavonoids have antibacterial properties [131,132]. Kil et al. (2009) reported that the antimicrobial properties of several medicinal plant extracts are due to the high quality of the flavonoids [133]. Flavonoids have a very large and diverse antibacterial activity, and they inhibit numerous microorganisms with different intensities depending on the pathogenic microorganism and the environment in which it is found; flavonoids are also able to prevent the growth of various microorganisms [134,135,136]. Therefore, the two flavonoids isolated from *C. odorata* could be used as markers of the antibacterial activities of this plant species.

#### 2.1.10. *Clerodendrum splendens* G.Don

*Clerodendrum splendens* (Lamiaceae (~7200 species in ~236 genera) is found in tropical Western Africa [137]. It is a climbing shrub that exhibits anti-inflammatory, antioxidant, antibacterial, and antifungal activities [138,139], and this has provided a scientific basis for its folkloric use in the treatment of numerous infectious conditions such as bruises, vaginal thrush, various skin infections and wound healing [138]. Preliminary phytochemical analysis of extracts showed the presence of unsaturated sterols, carbohydrates, glycosides, alkaloids, triterpenoids, tannins and flavonoids [140]. The methanol aerial extracts of *C. splendens* exhibited very good antifungal and antibacterial activities against the pathogenic bacteria tested with the minimum inhibitory concentrations (MIC) ranging from 64 to 256 μg/mL against *Candida albicans*, *Klebsiella pneumoniae*, *Proteus mirabilis*, *Staphylococcus aureus*, *Streptococcus faecalis*, *Bacillus subtilis*, *Escherichia coli*, and *Pseudomonas aeruginosa* using the micro-well dilution method [138]. Gbedema et al. (2010) reported that the methanol aerial extracts of *C. splendens* contained reducing sugars, phytosterols, tannins, terpenoids, alkaloids and flavonoids, which could be responsible for the high antifungal and antibacterial activities [138]. 

#### 2.1.11. *Combretum racemosum* P.Beauv.

*Combretum racemosum* belongs to the Combretaceae family, which consists of approximately ~600 species in ~20 genera [141]. The straggling shrub is widely spread across Africa to Senegal, Sudan, Nigeria, Kenya, Angola and the Democratic Republic of the Congo (ex-Zaïre) [141]. *C. racemosum* has been used for many years in African traditional therapeutic practices for the treatment of wounds, haemorrhoids, roundworms, gastro-intestinal affections, coughs, toothache, tuberculosis, genito-urinary, male sterility and bleeding during pregnancy [142,143,144,145]. Moreover, the antiulcer, vasorelaxant, anti-inflammatory, antimicrobial, antitrypanosomal, and antitrypanosomal effects of the *C. racemosum* extracts have been proven in modern biological studies [146,147,148,149]. Gossan et al. (2016) isolated betulinic acid, 28-*O*-β-D-glucopyranosyl-2α,3β,21β,23-tetrahydroxyolean-18-en-28-oate, 11 (3-*O*-β-acetyl-ursolic acid), terminolic acid, quadranoside, and arjungenin from the ethyl acetate extracts of *C. racemosum*. These metabolites showed moderate antibacterial activity against *Enterococcus faecalis*, *Escherichia coli* and *Staphylococcus aureus*, with minimum inhibitory concentrations that ranged from 64 to 256 μg/mL [141]. Therefore, these compounds were responsible for the antibacterial activities of ethyl acetate extracts of this plant species.

#### 2.1.12. *Dioscorea praehensilis* Benth.

*Dioscorea praehensilis* (Dioscoreaceae family with about ~715 species in ~9 genera) is a species of yam in the genus *Dioscorea* growing in tropical and subtropical areas of West Africa [150]. *D. praehensilis* is used by different ethnic groups and geographical areas to treat health problems, such as diabetes, stomach pains, rheumatism, haemorrhoids, coughs, skin infections, and diarrhoea [151,152]. Modern research has proven that *Dioscorea praehensilis* has a variety of biological activities, such as antidiabetic, anti-tumour, anti-inflammatory, and antibacterial activities [150]. The biological activities of *D. praehensilis* were mostly due to the presence of tannins, terpenoids, steroids, saponins, flavonoids, alkaloids, and anthocyanins [153]. Furthermore, Sautour et al. (2004) used mplc (medium-pressure liquid chromatography) column chromatography on silica gel to isolate 26-*O*-β-D-glucopyranosyl-22-methoxy-3β,26-di-hydroxy-25(R)-furost-5-en-3-*O*-α-L-rhamnopyranosyl-(1→4)-α-L-rhamnopyranosyl-(1→4)-[α-L-rhamnopyranosyl-(1→2)]-β-D-glucopyranoside; dioscin; and diosgenin 3-*O*-α-L-rhamnopyranosyl-(1→4)-α-L-rhamnopyranosyl-(1→4)-[α-L-rhamnopyranosyl-(1→2)]-β-D-glucopyranoside from the rhizomes of *D. praehensilis*. Dioscin showed antifungal activity against *Candida tropicalis* (MIC = 25.0 µg/mL), *Candida glabrata* (MIC = 12.5 µg/mL) and *Candida albicans* (MIC = 12.5 µg/mL) using the broth dilution test [150]. Diosgenin 3-*O*-α-L-rhamnopyranosyl-(1→4)-α-L-rhamnopyranosyl-(1→4)-[α-L-rhamnopyranosyl-(1→2)]-β-D-glucopyranoside had low activity; whereas 26-*O*-β-D-glucopyranosyl-22-methoxy-3β,26-di-hydroxy-25(R)-furost-5-en-3-*O*-α-L-rhamnopyranosyl-(1→4)-α-L-rhamnopyranosyl-(1→4)-[α-L-rhamnopyranosyl-(1→2)]-β-D-glucopyranoside was not active [150].

#### 2.1.13. *Erythrina abyssinica* Lam.

*Erythrina abyssinica* belongs to the plant family of the Fabaceae, with about 765 genera and nearly 20,000 known species ~765 genera and nearly ~20,000 known species [154,155,156]. The deciduous leguminous tree is found in Eastern DRC, Southern Africa and East Africa [157,158,159]. *E. abyssinica* is traditionally used to treat diseases such as malaria, tuberculosis, cancer, diabetes, leprosy, syphilis, back pain, yellow fever, anaemia, inflammatory diseases, venereal diseases, sexually transmitted diseases, skin infections, diarrhoea, epilepsy, urinary tract infections, pregnancy-related conditions, vomiting, soft tissue, hepatitis, central nervous system (CNS)-related disorders, helminthiasis, pneumonia, infertility, bacterial and fungal infections [25,160,161,162,163]. Preliminary phytochemical testing of several solvent extracts of *E. abyssinica* indicated the existence of flavonoids, phenols, terpenoids, chalcones, aromatic hydrocarbons, alkaloids, saponins, quinones and tannins as the major medicinal secondary metabolites [164,165]. Also, chemical compounds in this species possess antifungal, antioxidant, antimycobacterial, anti-HIV, anti-inflammatory, anticancer, antiviral, antihelmintic, antianemic, antibacterial, antiplasmodial, antiobesity, antidiabetic, antipyretic, and hepatoprotective bioactivities [166]. Antimicrobial activities of *E. abyssinica* crude extracts have been extensively assessed using the microbroth dilution assay against *Staphylococcus aureus* ATCC 25922 [167]. The existence of antimicrobial activities of *E. abyssinica* was attributed to 2′-methoxy-nor-glycyrrisoflavanone, 2′,3′,4′,7-tetrahydroxy-5′-prenylflavanone and 1,5,4′-trihydroxy-5′-prenylchalcone isolated from the plant. These pure compounds from *E. abyssinica* dichloromethane stem bark extracts were found active against *Staphylococcus aureus* (ATCC 25922) with a zone of inhibition of 15 mm at a concentration of 100 µg/mL [167]. Chitopoa et al. (2019) showed that the hexane, ethyl acetate, dichloromethane, ethanol, and methanol crude extracts of this plant had strong antifungal and antibacterial activities with MICs values of 3 and 10,000 μg/mL against different bacteria. Antimicrobial activity was noticed in most crude extracts, with the ethyl acetate crude extract showing the highest inhibition zone of 25 mm against *Candida albicans*, and the hexane and dichloromethane crude extracts were the most potent with minimum inhibitory concentrations of 62.5 μg/mL [168]. However, the hexane extract showed the highest inhibition zone of 23 mm against *Staphylococcus aureus*, while the dichloromethane was found to be the most active with a minimum inhibitory concentration of 15.6 μg/mL against *Candida albicans* [168]. Variation in the degree of activity in the *E. abyssinica* crude extracts might be due to the high quantity of terpenoids, which are known to possess antimicrobial, antifungal, and antibacterial activities [169,170,171].

#### 2.1.14. *Gardenia ternifolia* Schumach. & Thonn.

*Gardenia ternifolia* is a tree found in Angola, Senegal, Uganda, Ethiopia, Kenya, Botswana, Namibia, and Mozambique. It belongs to the Rubiaceae family, which consists of ~13,500 species in ~620 genera, and is one of the largest of the angiosperm family [172]. *G. ternifolia* is used to manage diabetes, malaria, sexually transmitted diseases, ascites, hepatitis, wounds, tooth decay, onchocerciasis, hypertension, haemorrhoids, leprosy, blood pressure, female infertility, skin diseases, diarrhoea, liver, cancer, sickle cell disease, rheumatism and yellow fever [25,173,174,175,176,177,178,179,180,181,182,183]. All parts of *G. ternifolia* are known to have antimicrobial, antitheilerial, antibiotic, antipain, anti-inflammatory, antisickling, antimalarial, antidiabetic, antileishmanial, larvicidal, antioxidant, hepatoprotective and cytotoxicity properties [184]. These pharmacological properties of *G. ternifolia* are attributed to saponins, steroids, alkaloids, terpenoids and polyphenols (tannins, coumarins and flavonoids) identified in different parts of *G. ternifolia* [174]. Ternifoliaoside A, β-sitosterol, β-stigmasterol-glucoside, 3β,23,24–trihydroxyurs-12-en-28-oic acid, 3β,19 α,23,24–tetrahydroxyurs-12-en-28-oic acid, lippianoside B, silphioside F, copteroside B and 24(24′)[Z]-dehydroamarasterone B were isolated from the roots of *Gardenia ternifolia* [185]. Ternifoliaoside A showed a very sensitive antibacterial effect on *Escherichia coli* (15.40 ± 2 mm) and *Salmonnela typhi* (20.1 ± 2 mm) at a concentration of 25 mg/mL using the Muller–Hinton agar diffusion method [185]. No effect was shown on *Escherichia coli*, *Vibrion cholorae*, *Salmonella typhi* and *Staphilococus aureaus* with compounds β-sitosterol and 24(24′)[Z]-dehydroamarasterone B at 25 mg/mL [185]. Furthermore, 3β,19 α,23,24–tetrahydroxyurs-12-en-28-oic acid exhibited the effect on *Staphylococcus aureus* (12.2 ± 2 mm) and *Pseudomonas aeruginosa* (15.6 ± 2 mm) at a concentration of 25 mg/mL [185].

#### 2.1.15. *Gloriosa superba* L.

*Gloriosa superba* is a flowering plant in the Colchicaceae family that includes ~285 species in ~15 genera. *G. superba* grows naturally throughout the sub-Saharan countries in the African continent to South Africa as well as in Madagascar [186,187]. It is also distributed in South Central China, Sri Lanka, Indonesia, and Southeast Asian countries. The alkaloid-rich plant has long been used in the treatment of wounds, gout, infertility, snakebite, cholera, ulcers, colic, arthritis, kidney problems, sprains, typhus, itching, smallpox, cancer, leprosy, sexually transmitted diseases, bruises, haemorrhoids, skin problems, nocturnal emission [188,189,190,191,192]. Ethnopharmacological studies have shown anthelmintic, antimicrobial, antibacterial, and antioxidant properties from *Gloriosa superba* [192]. Mustefa et al. (2024) showed that *G. superba* has antibacterial properties. Their results showed that 3-hydroxy-5-methoxy-benzoic acid, 3-hydroxymethyl phenol and desmosterol isolated from the chloroform and methanol extracts using silica gel column chromatography had potent in vitro antibacterial against *Klebsiella pneumoniae*, with inhibition zone values of (11.33 ± 1.15, 11.33 ± 1.53, and 12.33 ± 0.58, mm, respectively), at 1000 μg/mL. In addition, 3-hydroxy-5-methoxy-benzoic acid and desmosterol showed promising zone of inhibition against *Pseudomonas aeruginosa* strain (14 ± 1.00 and 14 ± 1.73 mm, respectively), at 100 μg/mL [193]. However, rutinose, sucrose, and 4-methoxy caffeic acid heptyl ester isolated from the same extracts did not display any activity [193].

#### 2.1.16. *Harungana madagascariensis* Lam. ex Poir.

*Harungana madagascariensis* belongs to the family Hypericaceae, comprising ~700 known species in ~9 genera. *H. madagascariensis* is distributed throughout the tropics and subtropics (Mauritius, Madagascar, and Africa growing on the edges of wet forests [194]. Furthermore, the plant is considered an antibiotic, anti-inflammatory, antimicrobial, anti-protozoan, anti-sickling, immunomodulatory, antioxidant, and enzyme inhibition effects, as well as cytotoxicity [195,196,197], and is used for the treatment of asthma, anaemia, tuberculosis, angina, syphilis, dysentery, gonorrhoea, parasitic skin diseases, hypertension, toothache, dysmenorrhea, hepatitis, ulcer, river blindness, malaria, diarrhoea, fever, yellow fever, chest pains and wounds [194,198,199]. Phytochemical investigation of *H. madagascariensis* has shown that the species contains metabolites such as anthracenic derivatives, saponins, tannins, flavonoids, alkaloids, and glycosides [195,196]. Moulari et al. (2006) showed that 3-*O*-α-L-rhamnoside-5,7,3′,4′-tetrahydroxydihydroflavonol (astilbin) contributed to the strong antibacterial activity against *Staphylococcus epidermidis*, *Micrococcus luteus*, *Moraxella* sp., and *Acinetobacter* sp., with minimal inhibitory quantity (MIQ) values of 50, 25, 50 and 50 µg, respectively [195]. Tankeo et al. (2016) isolated betulinic acid, ferruginin A, madagascin, and kaempferol-3-*O*-β-D-glucopyranoside from the methanol leaf and stem bark crude extracts of *H. madagascariensis* using silica gel column chromatography [200]. Ferruginin A had significant antibacterial activity, with MIC values below 10 µg/mL against *Enterobacter cloacae* (BM67), *Klebsiella pneumoniae* (K2 and Kp55), *Pseudomonas aeruginosa* (PA01), *Enterobacter aerogenes* (ATCC13048 and EA294), and *Escherichia coli* (ATCC10536) [200]. However, betulinic acid, madagascin and kaempferol-3-*O*-β-D-glucopyranoside showed poor inhibitory effects (MIC > 128 µg/mL) against all tested bacteria [200].

#### 2.1.17. *Hymenocardia acida* Tul.

*Hymenocardia acida* is a plant of the Phyllanthaceae family that is found in tropical Africa [201]. *H. acida* is used to treat chest complaints, diarrhoea, toothaches, smallpox, hypertension, headaches, rheumatic pains, abdominal tumours, jaundice, menstrual pains, abscesses, muscular pains and arthritis [202]. Phytochemical testing of the plant species showed the existence of carbohydrates, saponins, terpenes, glycosides, sterols, phenols, flavonoids and tannins [203]. Biological activities reported on the plant include antitumour, antiulcer, anti-HIV, antitrypanosomal, antiplasmodial, and antimicrobial, as well as cytotoxicity [204,205]. Six stilbenoid compounds (hymenocardichromene A–F) and one chromane stilbenoid (hymenocardichromanic acid) were isolated from the leaf extracts of *H. acida* using semipreparative HPLC [206]. Hymenocardchromanic acid showed the best antibacterial activity compared to the other compounds at 8 μg/mL against methicillin-resistant *Staphylococcus aureus* (MRSA-108), whereas the other compounds were moderately active [206]. Moreover, Agbidye et al. (2020) demonstrated that the methanol stem bark and root extracts of *H. acida* had potent antibacterial and fungal effects against *Staphylococcus aureus*, *Streptococcus pyogenes*, and *Candida albicans* at 1.0 × 10^3^ mg/mL, respectively [207]. However, none of the leaf extracts demonstrated antibacterial and antifungal effects against the tested pathogenic microorganisms [207]. Agbidye et al. (2020) further showed that ethyl isoallocholate, γ-sitosterol, β-sitosterol, lupeol, stigmasterol and friedelin tentatively identified by GC-MS spectral technique contributed to the strong antibacterial and fungal effects of the methanol stem bark and root extracts of *H. acida* [207].

#### 2.1.18. *Lannea edulis* (Sond.) Engl var.*edulis*.

*Lannea edulis* is a small deciduous shrub broadly used to treat malaria, gonorrhoea, angina pectoris, sexually transmitted diseases, dizziness, dysmenorrhea, schistosomiasis, bilharzia, gastrointestinal problems, diarrhoea, sore eyes, amenorrhea and to dilate the birth canal in Southern and East Africa [208,209,210,211,212,213,214]. *L. edulis* belongs to the Anacardiaceae family, comprising ~800 species in ~81 genera spread across tropical and subtropical regions in Africa, China, India, Indochina, and the Saudi Arabian Peninsula [215]. Ethnopharmacological studies showed that *Lannea edulis* extracts and constituents have antimalarial, antihyperglycemic, anti-human immunodeficiency virus, anthelmintic, antioxidant, antimicrobial, and antihyperlipidemic activities, as well as cytotoxicity [208,213,216,217,218,219,220,221]. In Malawi and South Africa, the seeds of *Vigna unguiculata* are mixed with the roots of *Lannea edulis* to treat blood urine and bilharzia [210,222,223]. However, in Malawi, the stem bark of *Piliostigma thonningii* is mixed with the stem bark of *Lannea edulis* to treat bilharzia [224]. The phytochemical screening of the genus *Lannea* showed that the species contains metabolites such as saponins, alkaloids, polyphenols, flavonoids, tannins, cardiac glycosides, and steroids from different plant parts of *L. edulis* [220,221]. Munodawafa et al. (2013) reported the antimicrobial properties of the leaf of *L. edulis* [218]. In their study, they found that the methanol crude extracts of *L. edulis* leaves were active against *Pseudomonas aeruginosa*, *Escherichia coli*, *Aspergillus niger*, *Staphylococcus aureus*, and *Candida albicans*, with MIC values that ranged from 2.5 to 5.0 mg/mL. The antimicrobial activity might be contributed to by the presence of alkaloids, coumarins, anthraquinone derivatives, cardiac glycosides, flavonoids, saponins and tannins detected in the methanol extracts of *L. edulis*. Medicinal plants that have flavonoids, alkaloids, and glycosides have been reported to possess antimicrobial activities [49].

#### 2.1.19. *Lippia multiflora* Moldenke

*Lippia multiflora* is a member of the Verbenaceae family, which includes about ~200 species of grasses, shrubs and small trees, usually found in an extensive ecological range throughout South and Central American countries and tropical West African regions [225,226]. Traditionally, *L. multiflora* is used for nausea, stomach aches, fevers, coughs, gastrointestinal disturbances, colds, enteritis, and as a laxative [227]. Various pharmacological activities have been reported for *L. multiflora,* among which are in vitro antimalarial, antiviral, antifungal, analgesic, antipyretic, anti-inflammatory, antimicrobial, and antifungal [228,229]. Phytochemical research on this plant has resulted in the isolation of carvacrol using an accelerated gradient chromatography technique with silica gel as an adsorbent [230]. Samba et al. (2020) demonstrated that volatile components present in *L. multiflora* have antibacterial effects against *Escherichia coli*, *Pseudomonas aeruginosa* and *Staphylococcus aureus* [226].

#### 2.1.20. *Morinda lucida* Benth.

*Morinda lucida* (“brimstone tree”) is a medicinal plant species in the Rubiaceae family which has been extensively used as medicine for decades in Central and West Africa for the treatment of conditions such as sickle cell disease, fever, cognitive disorders, typhoid fever, trypanosomiasis, malaria, parasitic worms, inflammation, cancer, hypertension, and diabetes [231,232]. Various compounds, including tannins, alkaloids, phenols, anthraquinones, fatty acids, saponins, flavonoids, anthraquinones, sterols, terpenoids, iridoids, polyphenols, and cardiac glycosides, have been isolated from different plant parts of *Morinda lucida* [233,234,235,236]. The in vitro scientific research studies on various extracts and pure components of *Morinda lucida* support the acclaimed biological properties of the plant, such as antioxidant, antimalarial, immunostimulatory, antimicrobial, anti-inflammatory, antidiabetic, hypotensive, anti-sickling, antileishmanial, antitrypanosomal, antifungal, antionchocercal, and antiproliferative [237]. Bata et al. (2023) proved that α and β-amyrin fractionated using column chromatography were responsible for the antibacterial activity of the active fraction of *M. lucida* methanol leaf against some multidrug-resistant Enterobacteriaceae (*Escherichia coli*, *Providencia* species, *Enterobacter*, *Serratia*, *Pragia*, and *Klebsiella*), with zones of inhibition that ranged from 15 to 18 mm [238].

#### 2.1.21. *Nauclea latifolia* Sm.

*Nauclea* is a tropical evergreen tree belonging to the Rubiaceae family. They are generally distributed in tropical areas in Africa [239]. Many African countries use *Nauclea latifolia* to treat fever, malaria, dental problems, ascites, toothaches, infectious diseases, hypertension, diarrhoea, dysentery, colic, epilepsy, wounds, hernia, vomiting and health promotion [240,241,242]. *N. latifolia* has such as anti-inflammatory, antioxidant, and antidiabetic activity [240,241,242]. *N. latifolia* contains phytochemicals such as carbohydrates, alkaloids, tannins, anthocyanins, saponins, phlobatannins, glycosides, terpenoids, and cardiac glycosides [239]. Chabi-Sika et al. (2022) assessed the antimicrobial activity of the ethanolic crude extract of *N. latifolia*. Their results showed that the extract had activity against *Streptococcus pneumoniae* (ATCC 496190) and *Pseudomonas aeruginosa* (ATCC 27853) with an MIC of 1.25 mg/mL and inhibition diameter of 19 ± 1.33 mm [243]. Phytochemical analysis revealed the presence of flavonoids, *O*-heterosides, C-heterosides, saponosides, anthocyanins, and mucilages [243]. These secondary metabolites, identified within the ethanolic crude extract of *N. latifolia*, are well known for their biological activities. Saponosides and flavonoids are recognised for their diversified antimicrobial, antibacterial, and antifungal activities.

#### 2.1.22. *Pachira glabra* Pasq.

The *Pachira glabra* is frequently distributed on wetlands near rivers and lakes. It belongs to the Malvaceae family and is native to the tropical regions of Africa, South America and Southern Mexico [244]. All parts of *P. glabra* are used to treat diarrhoea, stomach pain, and dysentery [245]. *Pachira glabra* is known to have antimicrobial, insecticidal, antimycobacterial, antioxidant and anti-helicobacter pylori activities [246,247]. These pharmacological properties are attributed to the presence of flavonoids, tannins, terpenoids, steroids, fatty acids, and coumarins in aerial and underground parts of *P. glabra* [248]. The antimicrobial activity of the *Pachira glabra* extracts using agar diffusion and broth microdilution methods has been reported, and the results showed that this plant has moderate to potent antimicrobial activity against *Micrococcus* spp., *Proteus* spp., and *Citrobacter youagae* with inhibition zones and MIC’s ranging from 13.7 to 24.0 mm and 0.3 to 2.5 mg/mL, respectively [249]. Thirty-three compounds with 98.40% of total contents from the essential oil of *P. glabra* contributed to the antimicrobial activity of the extracts [249].

#### 2.1.23. *Piliostigma thonningii* (Schum.) Milne-Redh.

*Piliostigma thonningii* is a leguminous plant that belongs to the Fabaceae family. It is used for numerous medicinal purposes in different African countries [250]. The decoction of the leaves and stem bark of *P. thonningii* are used for the management of malaria, ulcers, cough, sore throat, diarrhoea, bronchitis, toothache, leprosy, pyrexia, arthritis, heart pain, wounds, and gingivitis [251]. Its twigs and roots are used in the management of fever, dysentery, cough, skin diseases and wound infections [252]. The underground and aerial parts of *P. thonningii* are reported to possess antibacterial, anti-inflammatory, antilipidemic, and antihelminthic activities [253]. Antimicrobial evaluation of 50 % ethyl acetate leaf extracts from *P. thonningii* was active against *Escherichia coli*, *Staphylococcus epidermidis*, *Staphylococcus aureus*, and methicillin-resistant *Staphylococcus aureus* with MIC values below 500 µg/mL [254]. Akinpelu and Obuoto (2000) demonstrated that the methanol stem extract of *P. thonningii* had activity against *Staphylococcus aureus* (NCIB 8588), *Shigella dysenteriae* (LIO), *Serratia marcescens* (NCIB 1377), *Pseudomonas aeruginosa* (NCIB 950), *Proteus vulgaris* (NCIB 67), *Escherichia coli* (NCIB 86), *Corynebacterium pyogenes* (LIO), and *Bacillus subtilis* (NCIB 3610) at a concentration of 20 mg/mL [255]. Phytochemical studies on *P. thoningii* extracts showed the occurrence of distinct chemical components that accommodate the activities of this herbal plant. Among the identified chemical classes were diterpenes, saponins, volatile oils, terpenes, alkaloids, tannins, flavonoids, and carbohydrates [255].

#### 2.1.24. *Piper umbellatum* L.

*Piper umbellatum* (Piperaceae (~3600 species in ~5 genera) is broadly distributed in Brazil, Mexico, Bolivia, Peru, Central America, the West Indian Islands, and South America. The neotropical plant has also been introduced to Southeast Asia and Africa [256]. *P. umbellatum* is used in traditional medicine in various preparation forms for the treatment of colic, malaria, diarrhoea, dysentery, digestive problems, peptic ulcer, dyspepsia, pains, constipation, fever, intestinal parasites, stomach ache, urinary tract infections, bruises, wound healing, swelling, inflammation, rheumatism, and gastrointestinal diseases [257]. Phytochemical studies of *Piper umbellatum* have demonstrated the presence of terpenes, steroids, flavonoids and alkaloids [258]. Other studies have shown that crude extracts and isolated compounds derived from *Piper umbellatum* extracts contain a variety of biological activities, including anti-atherogenic, antibacterial, antimalarial, anti-inflammatory, analgesic, antioxidant, antifungal, anti-leishmanial, and antitrypanosomal, as well as cytotoxicity [259]. Okunrobo et al. (2011) demonstrated that carbohydrates, alkaloids, cardiac glycosides, tannins and saponins present in the methanol extract of *P. umbellatum*, as well as the n-hexane, chloroform and n-butanol fractions of the extract might be responsible for the antimicrobial activity against all the test pathogenic microorganisms (*Staphylococus aureus*, *Psuedomonas aeruginosa*, *Candida albicans* and *Escherichia coli*), with MIC values < 25 mg/mL [260].

#### 2.1.25. *Psorospermum febrifugum* Spach

*Psorospermum febrifugum* (Hypericaceae) is a plant species in the genus *Psorospermum* growing in Mozambique, Angola, Ethiopia, Zimbabwe and Guinea [261,262]. *P. febrifugum* has been used to treat various ailments, including epilepsy, skin diseases, wounds, scabies, eczema, pimples, leprosy, malaria, tuberculosis, pneumonia, poison, dysentery, dysmenorrhoea, whooping cough, skin rashes, syphilis, haemorrhoids, and stomach disorders; as well as opportunistic diseases such as watery blisters in genital areas, cryptococcal meningitis, and herpes [17,21,22,263,264,265]. Many phytochemicals, such as alkaloids, flavonoids, xanthones, steroids, anthraquinones, tannins, glycosides, terpenoids, and phenols, have also been isolated from the *Psorospermum febrifugum*, making it an important resource for biological drug and applications discovery [266,267,268]. Biological information from in vitro studies showed that *P. febrifugum* phytochemicals and extracts contain biological effects such as anticancer, anxiolytic, antipsoriatic, anti-inflammatory, antitrypanosomal, acaricidal, antimicrobial, antioxidant, and antimalarial, which affirmed the traditional usage of the plant [262,269,270,271,272,273]. Nambooze (2019) demonstrated that *P. febrifugum* had noteworthy antibacterial activity [274]. The ethyl acetate stem bark crude extract displayed the highest zone of inhibition of 18.3 ± 0.07 mm and 19.1 ± 0.14 mm against *Staphylococcus aureus* and *Streptococcus pyogenes*, respectively [274]. Phytochemical screening of ethyl acetate extract proved the existence of tannins, carbohydrates, reducing sugar, phenols, and terpenoids. Furthermore, Nambooze (2019) isolated, identified and characterised oleanolic acetate acid, betulinic acid and oleanolic acid from the ethyl acetate extract; a promising antibacterial activity was exhibited by all three metabolites [274]. Oleanolic acetate acid displayed the lowest zone of inhibitions of 8.1 ± 0.14 mm and 6.5 ± 0.2 mm against *Pseudomonas aeruginosa* and *Staphylococcus aureus*, whereas betulinic acid showed a moderate zone of inhibition [274]. Oleanolic acid displayed the highest zone of inhibition of (14.2 ± 0.07 mm) against *Staphylococcus aureus* and *Streptococcus pyogenes*. Additionally, Mpinda et al. (2018) proved that the extracts of *P. febrifugum* demonstrated growth inhibition of *Klebsiella pneumoniae* with the minimum inhibitory concentrations between 6.3 and 25 mg/mL [271]. This may be due to synergistic or additive effects of the compound mixtures in the extracts.

#### 2.1.26. *Syzygium guineense* Wall.

*Syzygium guineense* belongs to the Myrtaceae family, which contains approximately ~4000 species of trees and shrubs in ~140 genera. *S. guineense* is distributed in various parts of Africa [275]. The stem bark and root mixtures of *S. guineense* are used to manage stomach aches, diarrhoea, diabetes mellitus, and typhoid fever and are anthelmintic [276]. *Syzygium guineense* has been reported to possess antioxidant, anti-inflammatory, antifungal, and antibacterial activities [276]. Mavanza et al. (2023) showed that flavonoids, glycosides, terpenoids, phenolics saponins, steroids, alkaloids, quinones, and tannins identified in *S. guineense* extracts by phytochemical analysis could be responsible for the antibacterial properties observed in this plant against *Bacillus subtilis* (10.67 mm), *Escherichia coli* (14.33 mm), *Staphylococcus aureus* (15.00 mm) and *Salmonella enterica typhii* (9.33 mm) [277]. Antibacterial properties observed in *S. guineense* extract against *Bacillus subtilis*, *Escherichia coli*, *Staphylococcus aureus*, and *Salmonella enterica typhii* produced inhibition zones of 10.67, 14.33, 15.00, and 9.33 mm, respectively [277]. Qualitative analysis of chemical compounds of *S. guineense* extracts showed the presence of flavonoids, glycosides, terpenoids, phenolics saponins, steroids, alkaloids, quinones, and tannins [277]. Terminolic acid, 28-β-glucopyranosyl ester, oleanolic acid, arjunolic acid, 6-hydroxyasiatic acid, asiatic acid, and 2-hydroxyoleanolic acid were isolated from the *S. guineense* leaf extracts using column chromatography over silica gel [278]. Asiatic acid and terminolic acid showed significant antibacterial activity against *Escherichia coli*, *Bacillus subtilis* and *Shigella sonnei* with MICs of 3, 0.5, and 30 μg for asiatic acid and 5, 0.75, and 30 μg for terminolic acid. Arjunolic acid and 6-hydroxyasiatic acid were less active with MICs of 6, 3, and 50 μg against *Escherichia coli* and *Bacillus subtilis*, whereas oleanolic acid, 2-hydroxyursolic acid, arjunolic acid and 28-β-glucopyranosyl ester had no activity against *Escherichia coli* and *Bacillus subtilis* [278].

#### 2.1.27. *Vitex doniana* Sweet

*Vitex doniana* is a tree native to the Afrotropics (Sub-Saharan Africa including Madagascar), Southern Arabian Peninsula, and Western India. The *Vitex* genus belongs to the family Lamiaceae [279]. In folk medicine, the root, stem bark, leaf and fruit pulp are used to treat diseases such as malaria, jaundice, cancer, anaemia, malnutrition, dysentery, leprosy, gonorrhoea, diarrhoea, rickets, ancylostomiasis, backaches, gastrointestinal disorders, respiratory diseases fevers, stiffness, headache, rash, chickenpox, measles, hemiplegia fever, wounds, eye troubles, colic, burns, kidney troubles stomach complaints, liver diseases, leprosy, and to control bleeding after childbirth [25,280,281]. *Vitex doniana* has been reported to have antibacterial, anti-hepatotoxic, anti-dysentery, and antimalarial activities [280]. Several studies have reported that hydroxycinnamic acid, allicins saponins, terpenoids and flavonoids found in the crude extracts of *V. doniana* could be responsible for the antimicrobial effectiveness of *Vitex doniana* against pathogenic bacteria such as *Staphylococcus aureus*, *Escherichia coli*, *Shigella dysenteriae*, *Bacillus subtilis*, *Pseudomonas aeruginosa* and *Salmonella typhii* [282,283,284,285,286]. Sonibare et al. (2009) proved that 31.13% β-phellandrene, 28.3% phytol, and 12.6% β-caryophyllene found in the leaf essential oil of *V. doniana* have antimicrobial activity against *Proteus mirabilis*, *Bacillus subtilis* and *Candida albicans* [287]. In another study, Owolabi et al. (2022) demonstrated that 23.57% incensyl acetate, 16.87% phytol, 12.34% (E)-β-caryophyllene, and 9.73% phytone present in *V. doniana* have noteworthy antifungal effectiveness against *Aspergillus niger* with MIC of 78.1 μg/mL [288].

## 3. Materials and Methods

### 3.1. Selection of Published Articles

A literature search on medicinal plants used in the management of microbial infections in Angola was conducted by gathering information or data from books, journals, published M.Sc. and Ph.D. theses, as well as different electronic databases such as Wiley Online, ScienceDirect, Scopus, SciFinder, Web of Science, BMC, Elsevier, Google Scholar, SpringerLink, and PubMed. The following keywords, “Angola”, “Plants used for treating illnesses possibly related to infections”, “Plants used for treating infectious diseases”, “Plants used in treating infections/infectious diseases/microbial infections in Angola”, “Microbial infections”, “Infectious diseases ”, “Angola/Angolan plants”, “Angolan medicinal plants”, “Angolan traditional medicine”, “Angolan and antimicrobial plants”, “Antimicrobial/Antimicrobial plants”, “Angolan and antibacterial plants”, “Antibacterial/Antibacterial plants”, “Angolan and antifungal plants”, “Antifungal/Antifungal plants”, “Traditional/Traditional knowledge”, “Traditional medicine”, “Traditional medicinal plants”, “Medicinal plants”, “Medicinal herbs”, “Indigenous/Indigenous knowledge”, “Plants/Herbal/Medicine/Remedies”, “Ethnobotany/Ethnobotanical”, “Ethnopharmacology/Ethnopharmacological”, “Ethnomedicine/Ethnomedicinal”, and “Phytomedicine” were used to search for relevant articles.

### 3.2. Selection Criteria

The main inclusion criterion was published research articles related to the use of plants for managing diseases related to infections, with duration from January 1976 to November 2023. The scientific name of the plant had to be provided. The exclusion criteria were published studies or articles not written in the English language. Furthermore, published studies or articles that did not have information on Angolan medicinal plants used for treating infections, microbial infections and illnesses related to infections, and abstract-only accessed published articles. Four-step selection criteria were employed to identify the published articles included in this study. Step 1: the importance of studies was checked based on the titles and abstracts of the published articles. Step 2: titles and abstracts of the published articles were critically evaluated to match the inclusion criteria. The inclusion criteria were published studies or articles written in the English language, published studies or articles with information on Angolan medicinal plants used for treating infections, microbial infections and illnesses related to infections, and the provision of the scientific name of the medicinal plant. Step 3: the full text of the identified published papers obtained by the authors was assessed and screened based on the information or data acquired in the previous step (step 2) to make an informed decision as to whether to add or reject these publications in this review. Step 4: the remaining full-text published articles that met the inclusion criteria were thoroughly inspected. In total, 102 published articles were included in this study (Figure 2). The scientific names for the plants were validated using recognised databases, including Plants of the World Online (https://powo.science.kew.org/, accessed on 6 May 2024), The Plant List (http://www.plantlist.org/, accessed on 6 May 2024) and The World Flora Online (WFO) (http://www.worldfloraonline.org/, accessed on 6 May 2024) [289,290,291].

**Table 1 plants-13-02991-t001:** Angolan medicinal plants used for microbial and bacterial infections.

Species	Family	Part(s) Used	Traditional Uses	References
*Acanthus montanus* (Nees) T. Anderson	Acanthaceae	Root	Cystitis, pains, endometritis, urogenital infections, aches, leucorrhoea, urethral pain, urinary disease and furuncle	[43,44]
*Abelmoschus esculentus* (L.) Moench	Malvaceae	Leaves, fruits, seeds, pods	Depression, ulcers, sore throats, pulmonary inflammations and gastrointestinal irritations	[51,54]
*Aframomum alboviolaceum* (Ridl.) K. Schum.	Zingiberaceae	Leaves	Cure headache, cough, fever, amoebic dysentery, haemorrhoids, gastritis, myoma, pruritis, hypertension, filarial, malaria, pruritus, and gastritis	[57,58,59,60]
*Alchornea cordifolia* (Schumach. & Thonn.) Müll.Arg.	Euphorbiaceae	Whole plant	Venereal diseases, malaria, diarrhoea, wounds, cuts, sores, coughs, colds, eye problems, pigmentation problems, gastrointestinal, headaches, rheumatic pains, urinary disorders; as well as inflammatory disorders, fungal, parasitic, bacterial, parasitic, and disorders	[64,65,67,68,69,70]
*Aloe buettneri* A. Berger	Asphodelaceae	Root, leaves	Wounds, vitiligo, malaria, rheumatism, insect bites, burns and worm sores	[80]
*Annona stenophylla* Engl. & Diels	Annonaceae	Root, leaves	Coughs, wounds, helminthiasis, cancer, dysentery, diarrhoea, headache, asthma, fever, malaria, dermatitis, mental disorders, diabetes, and peptic ulcers	[87,88,89]
*Bridelia ferruginea* Benth.	Phyllanthaceae	Root, stem bark, leaves, fruits	Dysentery, arthritis, cough, rashes, constipation, epilepsy, diarrhoea, diuretic, chronic diabetes, asthma, gout, intestinal disorders, thrush, dysentery, gastralgias, rheumatisms, sexually transmitted diseases, contusion, oral infections, bladder disorders, skin diseases, and anaemia	[93,94]
*Canarium schweinfurtii* Engl.	Burseraceae	Stem bark, resin, leaves	Roundworm infections, skin-affections, dysentery, eczema, diarrhoea, dysentery, haemorrhoids, venereal diseases, hypertension, malaria, fever, gonorrhoea, chest pains, stomach complaints, pulmonary affections, ulcers, leprosy, coughs and wounds	[106,107,108,109]
*Chromolaena odorata* (L.) R.M.King & H.Rob.	Asteraceae	Leaves	Malaria, coughs, colds, toothache, stomach problems, diarrhoea, stomach ulcers, wounds, dysentery, skin infections, bacterial and fungal infections	[123,124,128,129]
*Clerodendrum splendens* G.Don	Lamiaceae	Whole plant	Bruises, vaginal thrush, various skin infections and wound healing	[137,138]
*Combretum racemosum* P.Beauv.	Fabaceae	Whole plant	Wounds, haemorrhoids, roundworms, gastro-intestinal affections, coughs, toothache, tuberculosis, genito-urinary, male sterility and bleeding during pregnancy	[141,142,143,144,145]
*Dioscorea praehensilis* Benth.	Dioscoreaceae	Whole plant	Diabetes, stomach pains, rheumatism, haemorrhoids, coughs, skin infections, and diarrhoea	[150,151,152]
*Erythrina abyssinica* Lam.	Fabaceae	Whole plant	Malaria, tuberculosis, cancer, diabetes, leprosy, syphilis, back pain, yellow fever, anaemia, inflammatory diseases, venereal diseases, sexually transmitted diseases, skin infections, diarrhoea, epilepsy, urinary tract infections, pregnancy-related conditions, vomiting, soft tissue, hepatitis, central nervous system (CNS)-related disorders, helminthiasis, pneumonia, infertility, bacterial and fungal infections	[25,154,155,156,160,161,162,163]
*Gardenia ternifolia* Schumach. & Thonn.	Rubiaceae	Whole plant	Diabetes, malaria, sexually transmitted diseases, ascites, hepatitis, wounds, tooth decay, onchocerciasis, hypertension, haemorrhoids, leprosy, blood pressure, female infertility, skin diseases, diarrhoea, liver, cancer, sickle cell disease, rheumatism and yellow fever	[25,172,173,174,175,176,177,178,179,180,181,182,183]
*Gloriosa superba* L.	Colchicaceae	Whole plant	Wounds, gout, infertility, snakebite, cholera, ulcers, colic, arthritis, kidney problems, sprains, typhus, itching, smallpox, cancer, leprosy, sexually transmitted diseases, bruises, haemorrhoids, skin problems, and nocturnal emission	[186,187,188,189,190,191,192]
*Harungana madagascariensis* Lam. ex Poir.	Hypericaceae	Stem bark, leaves	Asthma, anaemia, tuberculosis, angina, syphilis, dysentery, gonorrhoea, parasitic skin diseases, hypertension, toothache, dysmenorrhea, hepatitis, ulcer, river blindness, malaria, diarrhoea, fever, yellow fever, chest pains and wounds	[194,198,199]
*Hymenocardia acida* Tul.	Phyllanthaceae	Root, stem bark, leaves	Chest complaints, diarrhoea, toothaches, smallpox, hypertension, headaches, rheumatic pains, abdominal, tumours, jaundice, menstrual pains, abscesses, muscular pains and arthritis	[201,202]
*Lannea edulis* (Sond.) Engl. var. *edulis*	Anacardiaceae	Root, rootbark, stem bark, leaves	Malaria, gonorrhoea, angina pectoris, sexually transmitted diseases, dizziness, dysmenorrhea, schistosomiasis, bilharzia, gastrointestinal problems, diarrhoea, sore eyes, amenorrhea and to dilate birth canal	[208,209,210,211,212,213,214]
*Lippia multiflora* Moldenke	Verbenaceae	Leaves	Nausea, stomach aches, fevers, coughs, gastrointestinal disturbances, colds, and enteritis	[226,227]
*Morinda lucida* Benth.	Rubiaceae	Root, stem bark, leaves	Sickle cell disease, fever, cognitive disorders, typhoid fever, trypanosomiasis, malaria, parasitic worms, inflammation, cancer, hypertension, and diabetes	[231,232]
*Nauclea latifolia* Sm.	Rubiaceae	Whole plant	Fever, malaria, dental problems, ascites, toothaches, infectious diseases hypertension, diarrhoea, dysentery, colic, epilepsy, wounds, hernia, vomiting and health promotion	[239,240,241,242]
*Pachira glabra* Pasq.	Malvaceae	Stem bark, leaves, seeds	Diarrhoea, stomach pain, and dysentery	[244,245]
*Piliostigma thonningii* (Schum.) Milne-Redh.	Fabaceae	Root, stem bark, leaves	Malaria, ulcers, cough, sore throat, diarrhoea, bronchitis, toothache, leprosy, pyrexia, arthritis, heart pain, wounds, and gingivitis	[250,252]
*Piper umbellatum* L.	Piperaceae	Whole plant	Colic, malaria, diarrhoea, dysentery, digestive problems, peptic ulcer, dyspepsia, pains, constipation, fever, intestinal parasites, stomach ache, urinary tract infections, bruises, wound healing, swelling, inflammation, rheumatism, and gastrointestinal diseases	[256,257]
*Psorospermum febrifugum* Spach	Hypericaceae	Whole plant	Epilepsy, skin diseases, wounds, scabies, eczema, pimples, leprosy, malaria, tuberculosis, pneumonia, poison, dysentery, dysmenorrhoea, whooping cough, skin rashes, syphilis, haemorrhoids, and stomach disorders; as well as opportunistic diseases such as watery blisters in genital areas, cryptococcal meningitis, and herpes	[17,21,22,261,262,263,264,265]
*Syzygium guineense* Wall.	Myrtaceae	Stem bark, leaves, fruits, seeds	Stomach ache, diarrhoea, diabetes mellitus, and typhoid fever	[275,276]
*Vitex doniana* Sweet	Lamiaceae	Whole plant	Malaria, jaundice, cancer, anaemia, malnutrition, dysentery, leprosy, gonorrhoea, diarrhoea, rickets, ancylostomiasis, backaches, gastro-intestinal disorders, respiratory diseases fevers, stiffness, headache, rash, chickenpox, measles, hemiplegia fever, wounds, eye troubles, colic, burns, kidney troubles stomach complaints, liver diseases, leprosy, and to control bleeding after childbirth	[25,279,280,281]

## 4. Conclusions and Future Perspectives

This review focused on the use of medicinal plants in the management of microbial infections in Angola. A total of 27 plant species, representing 19 families, were documented in this study. The medicinal plants recorded in this study may assist researchers in developing novel medicines for the treatment of infectious diseases in Angola, on the African continent and beyond. This review provided data for scientific considerations in the search for novel natural-based drug development for the management of infectious diseases. There is a need to perform a comprehensive isolation of the bioactive compounds in the identified medicinal plants for biological analysis purposes and to determine their therapeutic properties. Furthermore, extensive preclinical and clinical trials are needed to investigate the therapeutic potential of medicinal plants for microbial infections. The employment of toxicological methods to validate the safety of the biologically active compounds isolated from the medicinal plants recorded or documented in this study is highly recommended.

## Figures and Tables

**Figure 1 plants-13-02991-f001:**
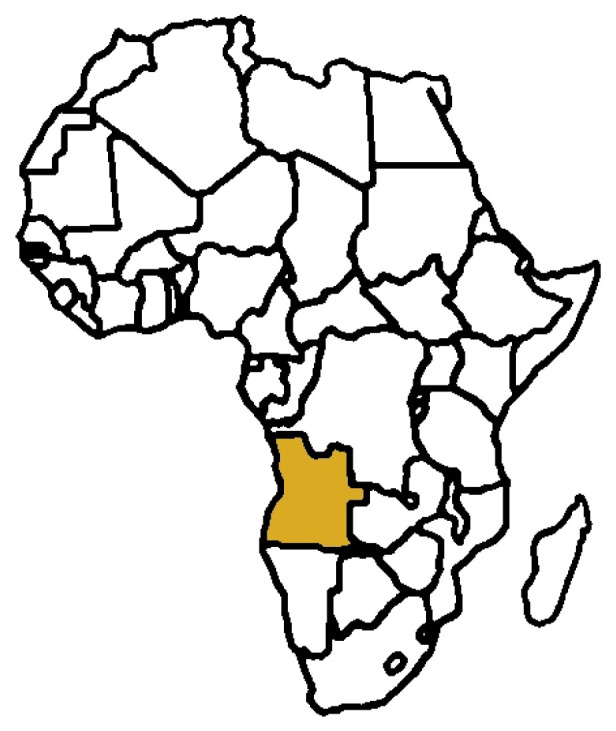
Angola’s geographic location in Africa (provided by Country Reports Org 2005 (https://www.countryreports.org/country/angola.htm), accessed on 4 March 2024).

**Figure 2 plants-13-02991-f002:**
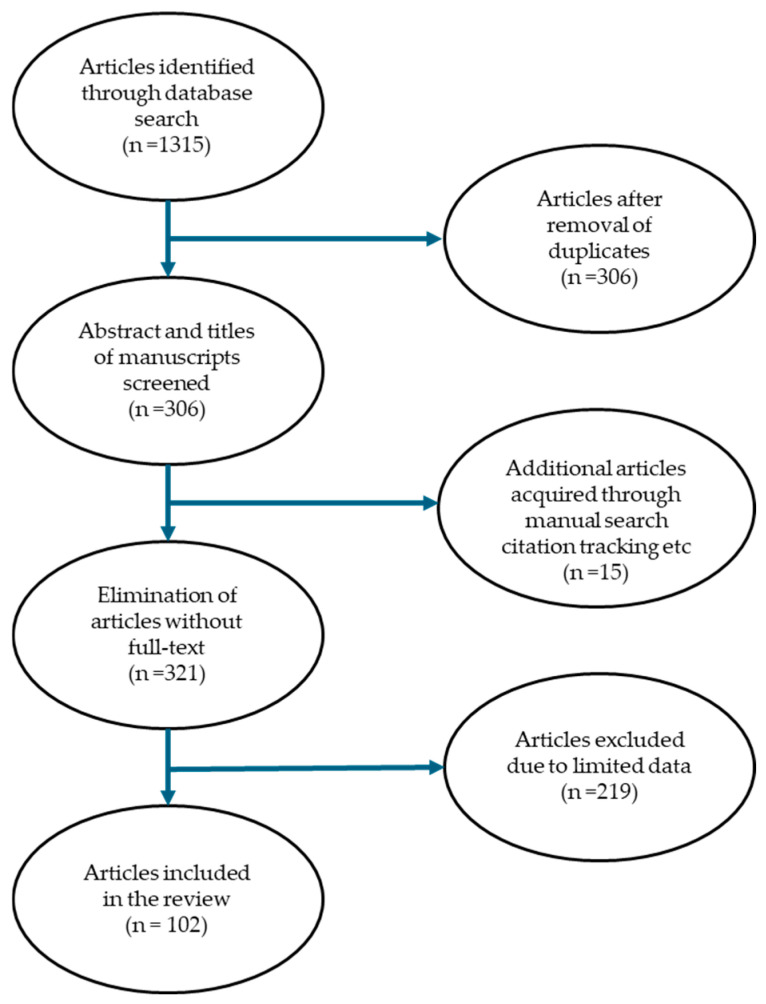
An overview of the procedure applied for the identification of 102 articles included in this review.

## Data Availability

Not applicable.

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
