# Peer review of "A Review of Medicinal Plants Used in the Management of Microbial Infections in Angola"

_plants, 2024, doi:10.3390/plants13212991_

Round 1

Reviewer 1 Report

Comments and Suggestions for Authors

The paper starts from a good idea but the methods adopted for conducting this review is not proper and therefore the whole paper is very problematic.

Authors just conducted a review based upon literature collected by using pharmacological key words - this means their review is about previous studies that explored the potential of Angolian medicinal plants already tested for a possible activities against infectious diseases. 

A serious traditional medicine centred review should start instead by looking at all the ethnobotanical sources that mention plants quoted against emic illnesses possibly related to infections.

The results of this very embarassing methodological mistake is a brief list of plants already well-known in the phytopharmacology.

The paper has no novelty and no value.

Moreover, authors were also not able to articulate the meaming of these poor data for future research or clinical applications.

Comments on the Quality of English Language

acceptable 

Reviewer 2 Report

Comments and Suggestions for Authors

The manuscript: A review of medicinal plants used in the management of microbial infections in Angola, described the state of the art of the use of medicinal plants from Angola in the treatment of infection diseases.

The paper is well understood, but there are some details that the authors did not see, some of them are:

i in page 11, line 499, the paragraph starts describing details of Abelmoschus esculenta, and then in the next description, section 2.118, the paragraph starts with the plant: Pachira glabra, line 517, but in the next line (518) the information is confuse because it refers to the previous plant. 

The authors should to correct such confusion.

Round 2

Reviewer 1 Report

Comments and Suggestions for Authors

The paper still has no value and is irrelevant, as already very clearly explained in my first review.

The authors still do not explain the criteria for the inclusion of the literature; there is no novelty assessment and no interpretation of the data.

Comments on the Quality of English Language

see above
